# To Optimize Radiotherapeutic Plans for Superior Tumor Coverage Predicts Malignant Glioma Prognosis and Normal Tissue Complication Probability

**DOI:** 10.3390/jcm11092413

**Published:** 2022-04-25

**Authors:** Chun-Yuan Kuo, Wei-Hsiu Liu, Yu-Ching Chou, Ming-Hsien Li, Jo-Ting Tsai, David YC Huang, Jang-Chun Lin

**Affiliations:** 1Department of Radiation Oncology, Shuang Ho Hospital, Taipei Medical University, Taipei 11031, Taiwan; 10637@s.tmu.edu.tw (C.-Y.K.); 09112@s.tmu.edu.tw (M.-H.L.); 10576@s.tmu.edu.tw (J.-T.T.); 2School of Biomedical Engineering, College of Biomedical Engineering, Taipei Medical University, Taipei 11031, Taiwan; 3Department of Neurological Surgery, Tri-Service General Hospital, National Defense Medical Center, No.325, Sec. 2, Cheng-Kung Road, Taipei 11490, Taiwan; doc20444@mail.ndmctsgh.edu.tw; 4Department of Surgery, School of Medicine, National Defense Medical Center, Taipei 11490, Taiwan; 5School of Public Health, National Defense Medical Center, Taipei 11490, Taiwan; trishow@mail.ndmctsgh.edu.tw; 6Department of Radiology, School of Medicine, College of Medicine, Taipei Medical University, Taipei 11031, Taiwan; 7Department of Medical Physics, Duke University, Durham, NC 27708, USA; yh126@duke.edu

**Keywords:** malignant glioma, IMRT, VMAT, TOMO, NTCP

## Abstract

Background: Radiotherapy (RT) provides a modern treatment to enhance the malignant glioma control rate. The purpose of our study was to determine the effect of tumor coverage on disease prognosis and to predict optimal RT plans to achieve a lower normal tissue complication probability (NTCP). Methods: Ten malignant-glioma patients with tumors adjacent to organs at risk (OARs) were collected. The patients were divided into two groups according to adequate coverage or not, and prognosis was analyzed. Then, using intensity-modulated radiation therapy (IMRT), volume-modulated arc therapy (VMAT), and helical tomotherapy (TOMO) to simulate new treatment plans for 10 patients, the advantages of these planning systems were revealed for subsequent prediction of NTCP. Results: The results of clinical analysis indicated that overall survival (*p* = 0.078) between the adequate and inadequate groups showed no differences, while the adequate group had better recurrence-free survival (*p* = 0.018) and progression-free survival (*p* = 0.009). TOMO had better CI (*p* < 0.001) and also predicted a lower total-irradiated dose to the normal brain (*p* = 0.001) and a lower NTCP (*p* = 0.027). Conclusions: The TOMO system provided optimal therapeutic planning, reducing NTCP and achieving better coverage. Combined with the clinical results, our findings suggest that TOMO can make malignant glioma patients close to OARs achieve better disease control.

## 1. Introduction

Malignant glioma is the most common and most aggressive type of brain cancer [1]. In addition, malignant glioma is characterized by a poor prognosis and high local recurrence rate. Recent studies have shown that the median time to local recurrence and progression of glioblastoma multiforme (GBM) is 7–9 months [2,3,4,5]. Currently, the standard treatment for GBM in clinical practice is to remove as much of the tumor as possible, followed by concurrent chemoradiotherapy (CCRT) after operation [6,7]. If radiotherapy (RT) is not started within 6 weeks after complete tumor resection, the survival rate of GBM patients is significantly reduced [8]. RT, as part of the main treatment course for patients with malignant gliomas, is an efficient and targeted treatment that can destroy possible malignant cells around brain tissues after surgical resection.

From clinical observation, the high-quality RT plan requires that the tumor coverage by the 100% prescribed dose is no less than 95% of the planning target volume (PTV). However, the tumor coverage must sometimes be limited to avoid damaging adjacent organs at risk (OARs), such as the brain stem, optic nerve, optic chiasm, lens, and cochlea/inner ear, reducing the quality of RT [9,10]. The closer the tumor is to the OARs, the more difficult it can be to determine the appropriate tumor coverage; insufficient tumor coverage can affect the patients’ prognosis, while tumor coverage that meets the requirement for adequate high-quality RT could result in the delivery of a high radiation dose to the OARs and subsequently could cause related complications and side effects. Moreover, because the life expectancy of GBM patients is significantly shortened, retaining neurological function and maintaining the ability to perform daily activities are important treatment goals [11,12,13].

Along with the evolution of RT technology, intensity-modulated radiation therapy (IMRT) has also been developed. Compared with three-dimensional conformal radiation therapy (3D-CRT), IMRT has better dose conformity, lower doses to OARs, and more rapid dose attenuation outside the target area while reaching similar or better tumor coverage, thereby better protecting the surrounding OARs and normal tissues [14,15,16,17,18]. Compared with IMRT, the greatest advantage of volume-modulated arc therapy (VMAT) is that the performance of VMAT is similar to that of IMRT but with a shorter dose delivery time, thereby improving treatment efficacy [19,20]. During VMAT, the rotational speed of the gantry, the output dose rate, and the shape of the treatment field, which can be adjusted by moving the leaves of the multileaf collimator (MLC), can be simultaneously changed [19,21]. Unlike IMRT and VMAT, which both use a linear accelerator (Linac) to deliver treatment, tomotherapy (TOMO) has a different mechanical configuration. TOMO applies helical radiation therapy by scanning with a fan beam emitted by a 6 MV single-photon energy Linac loaded on the circular gantry of a computed tomography (CT) scanner in a mode similar to CT scans, in combination with a treatment bed that can continuously move during treatment [22].

This study aimed to evaluate the effect of tumor coverage on malignant glioma response rate and the dosimetric differences among IMRT, VMAT, and TOMO for the treatment of malignant gliomas adjacent to OARs by comparing the tumor coverage and normal tissue complication probability (NTCP) of adjacent OARs.

## 2. Materials and Methods

### 2.1. The Patient Data and Simulation

A retrospective design was used to collect data from 10 patients with malignant glioma treated between December 2011 and November 2016, including 1 with oligoastrocytoma (OA), 3 with anaplastic astrocytoma (AA), and 6 with GBM. The clinical target volume (CTV) in the original computerized treatment plans of these 10 patients was close to at least 1 OAR, and in 9 patients, the tumors even caused different degrees of brain stem compression. The RT techniques were selected for these 10 patients based on the professional considerations of the attending physicians and the preferences of the patients. Five, two, and three cases were treated with IMRT, VMAT, and TOMO. The clinical condition of the 10 patients is listed in Table 1.

All 10 patients underwent CT simulation using the Philips Brilliance CT Big Bore at our hospital while in the supine position. The head was fixed with a thermoplastic head mask. The scanning range was from approximately 3 cm above the top of the head to approximately 5 cm below the foramen magnum at a slice thickness of 3 mm. CT images and diagnostic magnetic resonance imaging (MRI) images were input into the Pinnacle^3^ treatment planning system (Philips HealthCare, Fitchburg, MA, USA) of our hospital for image registration so that the attending physicians could delineate the target area of the tumor and the adjacent OARs. High-risk CTV (CTV_H60) was defined as the resection cavity and MRI T1-weighted enhancement zone after tumor resection. Low-risk CTV (CTV_L46) was defined as CTV_H60 plus the isometric margin resulting from the expansion of the CTV_H60 by 2 cm in each direction and included the peritumor edema zone. Then, the high-risk PTV (PTV_H60) and low-risk PTV (PTV_L46) were obtained by extending a 3-mm isometric margin from the CTV_H60 and CTV_L46, respectively. The adjacent OARs included the brain stem, optic nerve, optic chiasm, lens, cochlea/inner ear, normal brain, etc. The volume of the normal brain was defined as the remaining brain tissue after excluding the CTV_H60, brain stem, and optic chiasm from the whole brain.

### 2.2. Planning Target and Organ Constraints

Regardless of the original choices of RT techniques for these 10 cases, to compare the differences in the dosimetric performance of IMRT, VMAT, and TOMO, the same qualified medical physicist recreated the computerized treatment plans with the three RT techniques for the 10 cases in this study. Sequential boost, a commonly used clinical treatment method in our hospital, was used in all three computerized treatment plans. In phase I, a prescribed dose of 46 Gy was used to treat the PTV_L46 in 23 fractions. In phase II, a prescribed dose of 14 Gy was used to treat the PTV_H60 7 times for local dose enhancement. The clinical dose limits of adjacent OARs had to meet the requirements of the QUANTEC summary guideline [23]. In addition, the dose limit to each OAR in any stage of the computerized treatment plan had to be less than a certain proportion of the dose limit (depending on the ratio of the prescribed dose to the total dose at that stage). This conservative approach was applied to minimize the risk of complications in OARs so that the patients could have good quality of life after treatment. The PTV dose coverage requirements and the adjacent OAR clinical dose limit requirements in the three computerized treatment plans are listed in Table 2.

### 2.3. Radiotherapy Planning Technique

In the computerized treatment plans for all three treatment techniques, 6 MV photon beams were used. At least six coplanar treatment beams were used in all of the step-and-shoot IMRT treatment plans, and one noncoplanar treatment beam was selectively used in combination as needed. The incidence angle of the treatment beam was selected based on the principle that the treatment beam should be incidental from the affected (ipsilateral) side whenever possible, and the incidence from the far (contralateral) side should be avoided as much as possible to reduce the dose to normal brain tissue. Three or four partial coplanar arcs were used in all of the VMAT treatment plans. The incidence angle of the arc beam was selected based on the same principle used to select the incidence angle of the treatment beam in the IMRT treatment plans. The inverse treatment planning of IMRT and VMAT was optimized using the Pinnacle^3^ treatment planning system (Philips HealthCare, Fitchburg, MA, USA) of our hospital, and the dose calculation grid size all had a resolution of 3 mm. The Linac used for IMRT and VMAT in our hospital is an Elekta Synergy Linear accelerator (Elekta, Stockholm, Sweden). The MLC is equipped with 40 pairs of 1 cm wide leaves. The built-in computerized treatment plan system TomoTherapy Hi-Art software (version 5.1.6) (Accuray, Madison, WI, USA) was used for inverse treatment planning for helical TOMO. The three major setting parameters were the field width, pitch, and modulation factor of the fan beam. Based on comprehensive consideration of the requirements and the treatment times for the clinical treatment plans, the medical physicist set the field width to 1.05 cm, the pitch to 0.430, and the initial range of the modulation factor to 2.0–4.0 for all TOMO computerized treatment plans, and the actual range of the modulation factor was 1.714–3.788 after completion of the treatment plans. In addition, due to the different sizes of the fields of view of the CT images of the 10 cases, the resolutions of the dose calculation grid size of the TOMO treatment plans were also slightly different, ranging from 1.73 mm to 2.63 mm.

In this study, the dosimetric performances of IMRT, VMAT, and TOMO were compared by evaluating the quality of the computerized treatment plans for the three treatment techniques with several dosimetric parameters. In addition to the coverage of the PTV and the dose to adjacent OARs, the conformity index (CI), the gradient index (GI), the heterogeneity index (HI), and the NTCP of normal brain tissue were compared.

CI is defined as follows, according to the Paddick conformity index [24]:$$\mathrm{CI}=\frac{{\mathrm{TV}}_{\mathrm{PIV}}{}^{2}}{\mathrm{TV}\times \mathrm{PIV}}$$
where TV is the volume of the planned target, PIV is the volume covered by the prescribed dose, and TV_PIV_ is the volume of the part of the planned target covered by the prescribed dose. The closer the CI is to 1, the higher the dose conformity.

GI is defined as follows, according to the Paddick gradient index [25]:$$\mathrm{GI}=\frac{{\mathrm{PIV}}_{50\%\mathrm{PD}}}{\mathrm{PIV}}$$
where PIV_50%PD_ is the volume covered by 50% of the prescribed dose. GI represents the dose fall-off outside the TV, and the lower the GI, the better the protection of normal tissues.

The HI is defined as follows in this study [26]:$$\mathrm{HI}=\frac{\left(D_{\max}-D_{\min}\right)}{D_{\mathrm{mean}}}$$
where D_max_ is the maximum dose to the TV, D_min_ is the minimum dose to the TV, and D_mean_ is the average dose to the TV. The lower the HI, the more uniform the dose to the TV.

Because the tumors were adjacent to the OARs in the patients enrolled in this study, a conflict between the PTV coverage requirements and the adjacent OAR dose limit requirements could be expected. Subsequently, there must be a trade-off between the two, which is also a challenge of this study. Therefore, three scenarios were assumed to compare the advantages and disadvantages of IMRT, VMAT, and TOMO to identify planning benefits on tumor coverage, dose limitation of OARs, and NTCP: (1) When the three treatment techniques (IMRT_C, VMAT_C, and TOMO_C) all reached the adequate PTV coverage requirements, the doses to the adjacent OARs and the dosimetric parameters were compared. (2) When the three treatment techniques (IMRT_N, VMAT_N, and TOMO_N) all met the OAR dose limit requirements, the PTV coverage, no matter whether adequate or not, and the dosimetric parameters were compared. (3) For the same treatment technique, the effects of the adequate PTV coverage requirement and meeting the OAR dose limit requirement on the NTCP and prognosis were compared.

### 2.4. Statistical Analysis

All data were expressed as frequency, percentage, and mean ± SD. Continuous and categorical variables were compared between different PTV coverage groups (adequate or inadequate group) with the Mann–Whitney U test or the chi-square test (or Fisher’s exact test), as appropriate. In addition, we used the Kaplan-Meier method to estimate the recurrence-free survival (RFS), progression-free survival (PFS), and overall survival (OS) for patients. The log-rank test was used to evaluate differences in RFS, PFS, and OS between the different PTV coverage groups. Furthermore, we performed the Kruskal–Wallis test among different treatment types (IMRT, VMAT, and TOMO) on cancer tissue or normal tissue and the Mann–Whitney U Test between cancer tissue and normal tissue in the same treatment types for treatment parameters in phase I, phase II, and phase I + phase II. All statistical tests were two-sided, and a level of 0.05 was considered statistically significant. All data analyses were performed using SPSS version 23 (IBM SPSS Statistics 23).

## 3. Results

### 3.1. Patient Characteristics and Clinical Prognosis

We separated the 10 patients into two groups according to the PTV coverage. The adequate group included patients who had PTV coverage of more than 95% prescribed dose in the 95% targeted volume. There were four patients in the adequate group, including two men and two women with three frontal lobe tumors and one parietal lesion. In contrast, six patients were in the inadequate group, including two men and four women with four frontal brain lesions and two temporal tumors. Table 1 shows that there was no significant difference between the two groups in terms of age, gender, ECOG, tumor location, tumor side of brain, operation type, WHO grade, or chemotherapy administration.

Comparing the disease prognosis of malignant glioma between the two groups, there was no significant difference in overall survival (OS) (*p* = 0.078) between the adequate and inadequate groups, as shown in Figure 1. In contrast, better recurrence-free survival (RFS) (Figure 2a) and progression-free survival (PFS) (Figure 2b) were observed in the adequate coverage group, with statistically significant differences (*p* = 0.018 in RFS; *p* = 0.009 in PFS) compared to the inadequate group. These results suggest that optimal coverage of RT planning could have benefits in tumor local control. Therefore, we further arranged the dosimetric comparison of IMRT, VMAT, and TOMO to point out which planning can provide superior tumor coverage. It might be a hint that optimizing radiotherapeutic plans could make adequate tumor coverage to result in better RFS and PFS.

### 3.2. RT Simulated Planning

Furthermore, we arranged the three different planning systems, including IMRT, VMAT, and TOMO, to re-plan the therapeutic programs under two optimal conditions. First, regarding consistency, to optimize PTV coverage and achieve adequate conditions, IMRT_C, VMAT_C, and TOMO _C were set to V95% ≥ 95% PTV. However, in the first situation, OARs might receive an excessive dose, inducing more complications after RT. Thus, in the second situation, the possible PTV coverage was as large as possible with optimal dose limitations to OARs. IMRT_N, VMAT_N, and TOMO _N could achieve these conditions as Figure 3. Thus, we attempted to cross-compare the three planning systems with different optimal targets or the same planning system under different situations. Those results are shown in Table 3.

There was more optimal coverage with PTV and V100% in TOMO_C than with other RT planning (IMRT and VMAT) regardless of whether it was phase I or phase II planning (*p* = 0.002; *p* = 0.006). In phase I, TOMO_C planning had a lower PTV D_max_ (*p* = 0.01) and D_min_ (*p* = 0.001) and better CI (*p* < 0.001) and HI (*p* = 0.001) for the therapeutic programs. At the same time, TOMO_C could predict a lower total-irradiated dose to the normal brain (*p* = 0.001) and lower NTCP (*p* = 0.027), with significant differences observed compared to the first situation. However, VMAT_C had the lowest right lens dose, with significant differences observed when compared with TOMO_C and IMRT_C. Although TOMO_C could not achieve a better OAR dose than the other RT techniques under optimal coverage, the other OARs, except for right lens D_max_, were not significantly different among TOMO_C, VMAT_C, and IMRT_C.

In TOMO_N, we found a lower PTV D_min_ (*p* = 0.002) and PTV D_max_ (*p* = 0.009) and better CI (*p* = 0.001) and HI (*p* = 0.004) in phase I planning. There was also a better brain stem dose (*p* = 0.005) in phase I and a lower total normal brain dose (*p* = 0.004) under phase I + phase II for malignant glioma patients. There were no differences among the three RT planning systems in NTCP after optimal dosimetry to OARs.

In Table 4, we further analyze adequate planning with the three RT planning systems with optimal OARs or with the same planning system under different optimal situations. PTV_L46 was planned 46 Gy in 23 fractions in phase I planning, and following that, a prescribed dose of 14 Gy in 7 fractions was used to treat the PTV_H60 for local dose enhancement in phase II planning. Regardless of phase I or phase II planning under optimal dose limitations to OARs, there were no significant differences among the three RT planning systems, including IMRT_N, VMAT_N, and TOMO_N. In particular, the TOMO planning system could provide similar adequate PTV plans to both optimal targets, including V95% ≥ 95% PTV and OARs. However, when using the IMRT planning system to achieve optimal dose limitations to OARs, there was less adequate PTV coverage in IMRT_N than IMRT_C, showing a statistically significant difference (*p* = 0.003). VMAT_C also had more adequate planning for PTV coverage than VMAT_N (*p* = 0.033).

## 4. Discussion

RT is a critical modality of treatment for patients with malignant gliomas, including WHO grade III and grade IV gliomas. In our previous study, optimal tumor coverage could not always be attained with the RT technique because of the dose limitation to OARs. At the same time, we also demonstrated that, regardless of the higher or lower tumor coverage rates of the RT program, it had similar effects on the overall survival of GBM patients [27]. Following advances in delivery and precision, RT should continue to play an important role in realizing optimal disease outcomes for patients with malignant glioma [28]. Therefore, we attempted to classify subgroups of patients from our previous research [27]. We found that, due to inadequate planning, the recurrence-free survival and progression-free survival of malignant glioma patients with irradiated areas adjacent to vital organs were significantly different from those of patients with optimal PTV coverage.

Although overall survival was not affected by the tumor coverage rates of RT in our study, RT could be affected by treatments such as surgical interventions, chemotherapy and subsequent intensive care. According to Buckner et al., study in 2016 [29], the therapeutic benefit of CCRT seems to be better in IDH1-mutated oligoastrocytoma patients than RT alone. Therefore, multimodal treatments could improve overall survival in malignant glioma. A recent review article [30] has allowed an innovative tool, next-generation sequencing (NGS), to use liquid biopsy of glioma in the prediction of disease prognosis determining neural stem-like cells combined with different molecular markers to develop malignant glioma. These complicated modalities might induce differences in the disease survival rate and local tumor control. If there is a shorter time to tumor progression, patients should receive more medical interventions after the disease worsens.

A series of RT planning studies showed that TOMO often has greater benefit in terms of organ preservation for patients with glioma [31], breast cancer [32], gastric cancer [33], and rectal cancer [34]. Contouring of high-grade glioma for RT planning has shown that TOMO could provide superior OAR sparing compared with VMAT and IMRT techniques. Although HI and CI seem optimal with the TOMO system, PTV coverage consistently presented no significant differences among TOMO, VMAT, and IMRT. Compared with Liu et al.’s publication [31], we found not only superior CI but also better PTV coverage and PTV D_max_ with TOMO_C.

Increasing evidence has shown that RT side effects can be reduced by planning RT in advance with high precision. A retrospective study [35] of prostate cancer patients analyzed rectal toxicity and proved that VMAT was more effective in decreasing proctitis than IMRT adding topical medications. The pulmonary and cardiac radiation dose of left breast cancer can be reduced significantly via the deep-inspiration-breath-hold technique RT [36] and skin toxicity can be decreased through image-guided RT planning for breast cancer [32]. Modern RT technology advances effectively through VMAT to achieve nodule regression of Merkel cell carcinoma and prevent those patients’ skin toxicity [37]. Proton therapy can provide glioma patients with better sparing of healthy brain tissue from the irradiated field and preserve neurocognition [38]. We also identified the probability of normal brain complications with three RT planning systems. The Pinnacle3 treatment planning system (Philips HealthCare, Fitchburg, MA, USA) could provide the NTCP to determine the toxicity of TOMO, VMAT and IMRT. Compared with VMAT and IMRT, TOMO_C can significantly decrease the likelihood of complications in normal brain tissues. In other words, TOMO_C predicts similar NTCP results as TOMO_N, suggesting that the TOMO system can further optimize tumor coverage and spare OARs to prevent neuropathy. To the best of our knowledge, this study is the first to confirm the correlation between neurotoxicity and RT techniques.

## 5. Conclusions

Considered together, for patients with malignant glioma adjacent to OARs or that compresses the brain stem, the TOMO system could provide the optimal therapeutic plan to lessen the NTCP and obtain better coverage. In conclusion, our clinical results indicate that TOMO planning could allow malignant glioma patients requiring irradiation in areas adjacent to OARs (brain stem, optic chiasma, and inner ear). From clinical results, better coverage could make patients with malignant glioma have better RFS and PFS. Taken together, TOMO planning might have the trend to improve disease local control, and this conclusion should be proved after a randomized phase III study in the future.

## Figures and Tables

**Figure 1 jcm-11-02413-f001:**
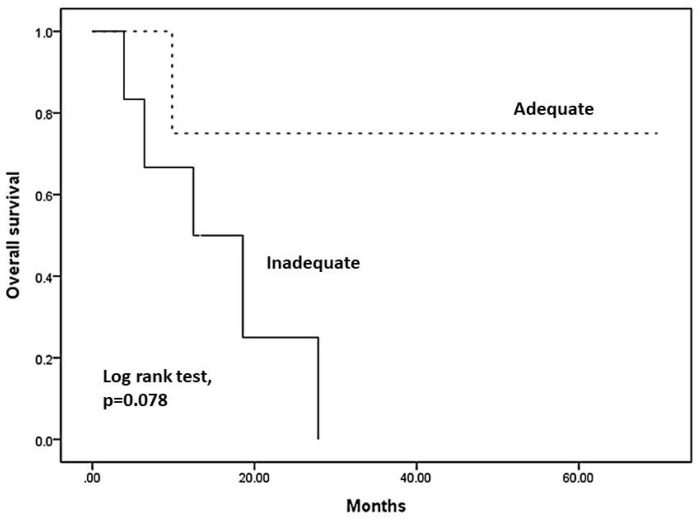
Kaplan–Meier overall survival by adequate group compared with inadequate (*p* = 0.078).

**Figure 2 jcm-11-02413-f002:**
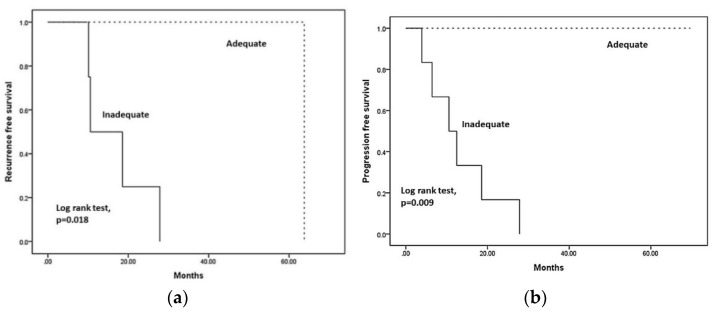
(**a**) Recurrence-free survival for comparing adequate with inadequate group (*p* = 0.018); (**b**) progression-free survival for comparing adequate with inadequate group (*p* = 0.009).

**Figure 3 jcm-11-02413-f003:**
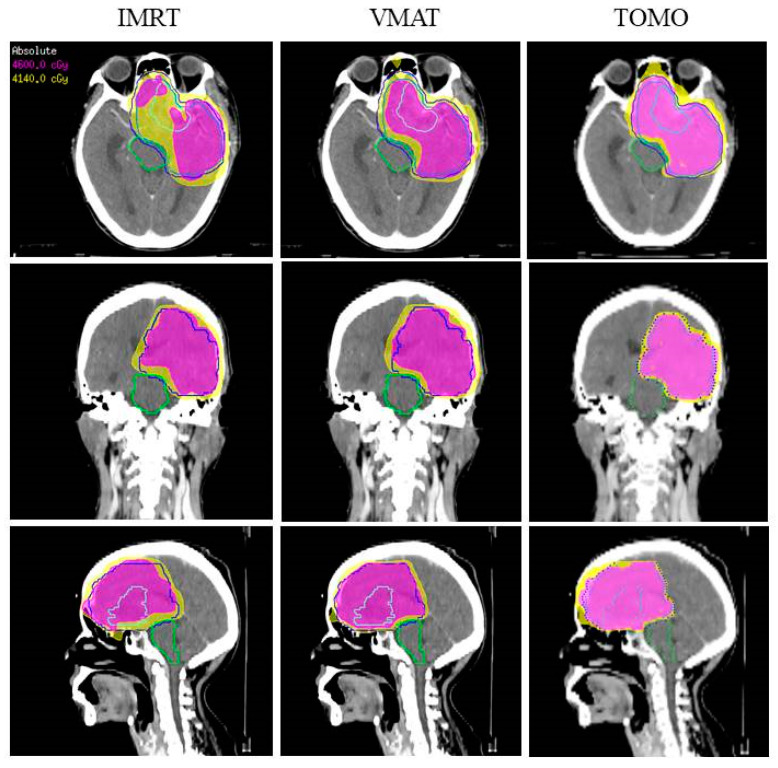
The 4th patient’s dose distributions on IMRT, VMAT, and TOMO. Color-wash areas: 46.00 Gy (pink), 41.40 Gy (yellow). IMRT: intensity-modulated radiation therapy; VMAT: volumetric-modulated arc therapy; TOMO: helical tomotherapy.

**Table 1 jcm-11-02413-t001:** Patients and tumor characteristics (*n* = 10).

PTV Coverage			
Variables	Adequate (*n* = 4)	Inadequate (*n* = 6)	*p* Value
**Age**	56.5 ± 22.31	54.5 ± 9.07	0.874
**Gender**			1.000
Female	2 (50.0)	4 (66.7)	
Male	2 (50.0)	2 (33.3)	
**ECOG ^1^**			1.000
0	1 (25.0)	1 (16.7)	
1	2 (50.0)	4 (66.7)	
2	1 (25.0)	1 (16.7)	
**Tumor location**			0.467
Frontal lobe	3 (75.0)	4 (66.7)	
Parietal lobe	1 (25.0)	0 (0)	
Occipital lobe	0 (0)	0 (0)	
Temporal lobe	0 (0)	2 (33.3)	
**Tumor side of brain**			1.000
Right side	2 (50.0)	4 (66.7)	
Left side	2 (50.0)	2 (33.3)	
Bilateral	0 (0)	0 (0)	
**Operation**			0.667
Total resection	2 (50.0)	5 (83.3)	
Subtotal resection	1 (25.0)	0 (0)	
Biopsy only	1 (25.0)	1 (16.7)	
**WHO Grade ^2^**			0.432
OA	0 (0)	1 (16.7)	
AA	1 (25)	2 (33.3)	
GBM	3 (75)	3 (50)	
**Chemotherapy**			1.000
No	0 (0)	1 (16.7)	
Yes (Temozolomide)	4 (100)	5 (83.3)	

^1^ ECOG: Eastern Cooperative Oncology Group. ^2^ OA: oligoastrocytoma; AA: anaplastic astrocytoma; GBM: glioblastoma multiforme.

**Table 2 jcm-11-02413-t002:** Target coverage goal and dose constraints for the critical organ.

Structure	PTV ^1^ Coverage Goal/OARs Dose Constraints
**Targets**	
PTV Coverage	V100% ≥ 95% PTV, Ideally V95% ≥ 95% PTV, Adequate
PTV maximum dose	<110% prescribed dose
**OARs** ^2^	
Brain stem	D_max_ ≤ 54 Gy
Lens	D_max_ ≤ 5 Gy
Optic nerve/chiasm	D_max_ ≤ 55 Gy
Cochlea/Inner ear	D_mean_ ≤ 45 Gy

^1^ PTV: planning target volume. ^2^ OARs: organs at risk.

**Table 3 jcm-11-02413-t003:** Dosimetric results for planning target volume and organs at risk in glioblastoma.

Variable	IMRT		VMAT		TOMO		C v N	C v N	C v N	IMRT v VMAT v TOMO	
	Mean + SD						Quick Reference Guide (*p-*Value) *				
	C	N	C	N	C	N	IMRT	VMAT	TOMO	C	N
**Tumor volume (cm^3^)**											
Phase I	482.13 ± 156.79	482.13 ± 156.79	482.13 ± 156.79	482.13 ± 156.79	482.13 ± 156.79	482.13 ± 156.79					
Phase II	203.32 ± 139.56	203.32 ± 139.56	203.32 ± 139.56	203.32 ± 139.56	203.32 ± 139.56	203.32 ± 139.56					
**PTV Coverage V100% (%)**											
Phase I	95.07 ± 0.08	89.68 ± 5.51	95.04 ± 0.08	91.75 ± 4.09	95.30 ± 0.25	93.93 ± 2.56	**0.013 ***	**0.031 ***	0.126	**0.002 ***	0.099
Phase II	95.17 ± 0.17	93.59 ± 4.36	95.08 ± 0.08	93.32 ± 3.72	95.38 ± 0.29	94.77 ± 1.98	0.280	0.170	0.341	**0.006 ***	0.624
**PTV Coverage V95% (%)**											
Phase I	98.87 ± 0.76	95.66 ± 3.31	98.75 ± 1.01	96.45 ± 2.85	98.89 ± 0.94	97.92 ± 2.23	**0.014 ***	**0.027 ***	0.216	0.931	0.214
Phase II	99.45 ± 0.54	98.42 ± 2.94	99.47 ± 0.50	98.51 ± 2.47	99.72 ± 0.23	99.23 ± 1.62	0.289	0.241	0.351	0.335	0.716
**PTV D_max_ (Gy)**											
Phase I	50.21 ± 0.54	50.28 ± 0.38	49.38 ± 0.69	49.65 ± 0.40	49.37 ± 0.72	49.48 ± 0.80	0.751	0.301	0.750	**0.010 ***	**0.009 ***
Phase II	15.02 ± 0.30	15.00 ± 0.29	14.90 ± 0.24	14.93 ± 0.28	14.69 ± 0.24	14.75 ± 0.29	0.875	0.777	0.631	**0.029 ***	0.158
**PTV D_min_ (Gy)**											
Phase I	37.02 ± 2.21	34.05 ± 2.80	36.82 ± 2.90	34.98 ± 3.58	31.25 ± 4.30	29.19 ± 3.98	0.017 *	0.223	0.280	**0.001 ***	**0.002 ***
Phase II	11.92 ± 1.46	11.63 ± 1.59	11.79 ± 1.65	11.17 ± 1.79	12.14 ± 0.93	11.64 ± 1.85	0.684	0.433	0.456	0.849	0.790
**Planning CI**											
Phase I	0.83 ± 0.04	0.80 ± 0.06	0.87 ± 0.03	0.84 ± 0.05	0.92 ± 0.03	0.90 ± 0.04	0.270	0.162	0.349	**<0.001 ***	**0.001 ***
Phase II	0.81 ± 0.05	0.80 ± 0.08	0.82 ± 0.08	0.81 ± 0.09	0.93 ± 0.02	0.93 ± 0.03	0.631	0.798	0.905	**<0.001 ***	**<0.001 ***
**Planning GI**											
Phase I	2.60 ± 0.43	2.72 ± 0.53	2.59 ± 0.48	2.66 ± 0.48	2.60 ± 0.37	2.62 ± 0.40	0.569	0.746	0.929	0.999	0.879
Phase II	3.82 ± 1.26	3.88 ± 1.29	4.00 ± 1.69	4.22 ± 2.24	3.54 ± 0.76	3.62 ± 0.90	0.915	0.812	0.852	0.729	0.698
**Planning HI**											
Phase I	0.28 ± 0.05	0.34 ± 0.06	0.26 ± 0.06	0.31 ± 0.08	0.39 ± 0.09	0.44 ± 0.09	0.016 *	0.163	0.266	**0.001 ***	**0.004 ***
Phase II	0.22 ± 0.12	0.23 ± 0.12	0.20 ± 0.08	0.24 ± 0.10	0.18 ± 0.08	0.22 ± 0.15	0.717	0.276	0.468	0.710	0.927
**Brain stem** **D_max_ (Gy)**											
Phase I	44.38 ± 2.69	41.32 ± 0.10	43.83 ± 2.96	41.20 ± 0.38	42.82 ± 3.16	40.80 ± 0.44	**0.006 ***	0.020 *	0.075	0.492	**0.005 ***
Phase II	10.07 ± 4.87	9.78 ± 4.65	9.88 ± 4.61	9.47 ± 4.29	9.15 ± 4.90	8.99 ± 4.74	0.893	0.837	0.942	0.903	0.927
Phase I + II	54.17 ± 5.41	51.00 ± 4.62	53.50 ± 5.47	50.59 ± 4.09	51.58 ± 5.39	49.23 ± 4.24	0.175	0.195	0.294	0.548	0.639
**Optic chiasm** **D_max_ (Gy)**											
Phase I	40.95 ± 7.87	38.65 ± 6.79	39.34 ± 10.08	37.71 ± 9.08	37.90 ± 11.20	36.42 ± 10.15	0.493	0.709	0.761	0.787	0.851
Phase II	8.64 ± 5.79	8.36 ± 5.54	8.24 ± 6.14	7.97 ± 5.89	7.95 ± 5.59	7.82 ± 5.45	0.914	0.919	0.959	0.966	0.976
Phase I + II	49.46 ± 12.07	46.83 ± 10.90	47.47 ± 14.62	45.48 ± 13.49	45.55 ± 14.35	44.02 ± 13.69	0.615	0.756	0.810	0.817	0.886
**Left optic nerve D_max_(Gy**)											
Phase I	29.38 ± 15.39	28.42 ± 15.32	28.51 ± 15.83	28.16 ± 15.73	27.09 ± 15.08	26.10 ± 14.79	0.890	0.962	0.883	0.945	0.933
Phase II	6.44 ± 5.19	6.33 ± 5.32	6.63 ± 5.51	6.61 ± 5.49	5.45 ± 4.56	5.34 ± 4.57	0.962	0.994	0.955	0.858	0.845
Phase I + II	35.72 ± 19.92	34.68 ± 20.05	34.97 ± 20.43	34.68 ± 20.49	32.01 ± 18.38	30.88 ± 18.10	0.909	0.975	0.891	0.905	0.882
**Right optic nerve D_max_ (Gy)**											
Phase I	31.05 ± 14.89	30.08 ± 14.11	27.96 ± 15.11	26.92 ± 14.12	26.39 ± 14.20	25.84 ± 13.88	0.882	0.875	0.932	0.773	0.784
Phase II	5.59 ± 4.37	5.57 ± 4.36	5.24 ± 4.56	5.00 ± 4.19	4.36 ± 3.42	4.52 ± 3.67	0.993	0.903	0.920	0.794	0.848
Phase I + II	36.54 ± 17.97	35.60 ± 17.06	33.17 ± 18.39	31.89 ± 17.02	30.56 ± 16.41	30.17 ± 16.32	0.906	0.873	0.959	0.750	0.763
**Left lens** **D_max_ (Gy)**											
Phase I	3.93 ± 1.34	3.37 ± 0.84	2.88 ± 0.98	2.87 ± 0.82	2.95 ± 1.00	2.81 ± 0.92	0.276	0.988	0.752	0.082	0.301
Phase II	0.77 ± 0.41	0.76 ± 0.41	0.62 ± 0.34	0.63 ± 0.34	0.62 ± 0.37	0.62 ± 0.37	0.953	0.995	0.976	0.631	0.663
Phase I + II	4.68 ± 1.63	4.11 ± 1.12	3.49 ± 1.14	3.48 ± 1.03	3.55 ± 1.31	3.41 ± 1.22	0.377	0.985	0.802	0.114	0.326
**Right lens** **D_max_ (Gy)**											
Phase I	3.77 ± 0.86	3.45 ± 0.69	2.86 ± 0.74	2.82 ± 0.73	3.05 ± 0.80	3.11 ± 0.83	0.367	0.895	0.869	**0.041 ***	0.192
Phase II	0.78 ± 0.41	0.78 ± 0.41	0.64 ± 0.34	0.65 ± 0.34	0.65 ± 0.38	0.64 ± 0.37	0.996	0.964	0.981	0.656	0.672
Phase I + II	4.53 ± 1.16	4.21 ± 0.97	3.49 ± 0.95	3.46 ± 0.94	3.65 ± 1.04	3.72 ± 1.06	0.519	0.931	0.890	0.079	0.240
**Left inner ear** **D_mean_ (Gy)**											
Phase I	20.16 ± 14.18	17.11 ± 12.72	16.26 ± 11.69	16.43 ± 11.71	14.66 ± 11.20	14.23 ± 11.17	0.619	0.974	0.932	0.602	0.852
Phase II	2.43 ± 2.86	2.58 ± 2.89	2.45 ± 2.40	2.30 ± 2.36	1.58 ± 1.74	1.57 ± 1.75	0.907	0.890	0.991	0.655	0.627
Phase I + II	22.58 ± 16.07	19.69 ± 14.65	18.71 ± 13.51	18.73 ± 13.56	16.17 ± 12.31	15.73 ± 12.32	0.679	0.997	0.938	0.596	0.794
**Right inner ear D_mean_ (Gy)**											
Phase I	19.11 ± 15.01	17.80 ± 12.73	17.78 ± 13.87	16.37 ± 12.61	14.49 ± 14.68	13.40 ± 12.72	0.836	0.815	0.862	0.767	0.734
Phase II	2.34 ± 2.84	2.60 ± 3.26	2.86 ± 3.50	2.70 ± 3.27	1.75 ± 2.27	1.69 ± 2.25	0.851	0.919	0.956	0.699	0.706
Phase I + II	21.45 ± 16.57	20.41 ± 14.97	20.64 ± 16.09	19.07 ± 14.76	16.21 ± 15.70	15.07 ± 13.75	0.884	0.823	0.865	0.740	0.697
**Normal brain (WB-CTV_H)**											
Phase I + II D_max_ (Gy)	64.44 ± 0.80	64.29 ± 0.74	63.34 ± 0.82	63.52 ± 0.61	62.60 ± 1.15	62.83 ± 1.17	0.669	0.574	0.656	**0.001 ***	**0.004 ***
Phase I + II D_mean_ (Gy)	34.80 ± 4.12	34.35 ± 3.89	33.67 ± 3.59	33.01 ± 3.33	31.85 ± 3.24	31.67 ± 3.20	0.804	0.674	0.900	0.213	0.247
**Phase I + II** **NTCP (%)**	**5.20 ± 2.57**	4.50 ± 2.37	**4.20 ± 1.75**	4.00 ± 1.70	2.70 ± 1.34	**2.70 ± 1.34**	0.535	0.798	1.000	**0.027 ***	0.697

IMRT, intensity-modulated radiation therapy; VMAT, volumetric-modulated radiation therapy; TOMO, tomotherapy; SD, standard deviation; V_X_, the percentage of organ receiving more or equal to x Gy; D_max_, maximum dose of certain OAR; D_mean_, mean dose of certain OAR. * Quick reference guide is based on the significant *p*-value (*p* < 0.05).

**Table 4 jcm-11-02413-t004:** Comparison of PTV coverage through different planning technique compassion.

	PTV Coverage		
Variables	Adequate	Inadequate	*p*-Value
**Phase I**			0.202
IMRT_N	3	7	
VMAT_N	5	5	
TOMO_N	7	3	
**Phase II**			1.000
IMRT_N	8	2	
VMAT_N	8	2	
TOMO_N	9	1	
**Phase I**			**0.003 ***
IMRT_C	10	0	
IMRT_N	3	7	
**Phase II**			0.474
IMRT_C	10	0	
IMRT_N	8	2	
**Phase I**			**0.033 ***
VMAT_C	10	0	
VMAT_N	5	5	
**Phase II**			0.474
VMAT_C	10	0	
VMAT_N	8	2	
**Phase I**			0.211
TOMO_C	10	0	
TOMO_N	7	3	
**Phase II**			1.000
TOMO_C	10	0	
TOMO_N	9	1	

* Quick reference guide is based on the significant *p*-value (*p* < 0.05).

## Data Availability

Not applicable due to partial ongoing study of data.

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
