# Peer review of "To Optimize Radiotherapeutic Plans for Superior Tumor Coverage Predicts Malignant Glioma Prognosis and Normal Tissue Complication Probability"

_jcm, 2022, doi:10.3390/jcm11092413_

Round 1

Reviewer 1 Report

The article is interesting because often in clinical practice and planning,
in particular for localized lesions very close to the OARs,
less coverage of the PTV is accepted, to avoid neurotoxicity (e.g. brain stem, optic pathway). 

In this study the dosimetric aspects and tumor coverage in the three different techniques IMRT, VMAT and TOMO are compared.

line 187: the three scenarios assumed to comapre IMRT, VMAT and TOMO are not explained very clearly. They should be better explained in detail.

line 211: Furthermore, I could not understand the division in adequate and inadequate group according to PTV coverage.

line 268: it would be useful to explain in detail what phase I and phase II consist of .

Author Response

Dear Reviewer 1:

  1. line 187: the three scenarios assumed to compare IMRT, VMAT, and TOMO are not explained very clearly. They should be better explained in detail.

Authors’ Response: Thank you for your comments and suggestions. We have edited the introduction for explaining the different planning systems among IMRT, VMAT, and TOMO. We also write the purpose of 3 assumptions with red marked words.

  1. line 211: Furthermore, I could not understand the division into adequate and inadequate groups according to PTV coverage.

Authors’ Response: Thank you for your question. As the guideline evaluates the quality of those RT planning, references according to “International Commission on Radiation Units and Measurements ICRU Report No 62: Prescribing, Recording and Reporting Photon Beam Therapy (Supplement to ICRU Report 50). Bethesda, MD: ICRU Publications 1999” would make the rule for coverage the high-quality RT plan requires that the tumor coverage by the 100% prescribed dose is no less than 95% of the planning target volume (PTV). However, in clinical practice, it is too hard to achieve these constraints. Therefore, in our Hospital, it could accept V95% ≥ 95% PTV (95% volume of PTV receiving more or equal to 95% planning RT dose) as the adequate quality of treatment planning after making discussion by those radiation oncologists. We hope to resolve your problem under our explanation.

  1. line 268: it would be useful to explain in detail what phase I and phase II consist of.

Authors’ Response: Thank you for your comment. From material and method, we had described that “In phase, I, a prescribed dose of 46 Gy was used to treat the PTV_L46 in 23 fractions. In phase II, a prescribed dose of 14 Gy was used to treat the PTV_H60 7 times for local dose enhancement.”  We also write “PTV_L46 would be planed 46Gy in 23 fractions in phase I planning and following, a prescribed dose of 14 Gy in 7 fractions was used to treat the PTV_H60 for local dose enhancement in phase II planning.” with red marked words to further explain the phase I and phase II  consist of.

Reviewer 2 Report

Manuscript Number: JCM-1646563: To Optimize Radiotherapeutic Plans for Superior Tumor Coverage in Predicts Malignant Glioma Prognosis and Normal  Tissue Complication Probability

The authors performed a DVH and NTCP-based dosimetric comparison on a small group of 10 patients with malignant glioma with three different techniques (IMRT, VMAT, Tomotherapy), for which the treatment plans have been recalculated.

First of all, the English needs to be thoroughly revised. The title is incomprehensible and this reviewer struggled to understand even the abstract.

The introduction is too long and needs to be shortened. There are generalities and even misconceptions (for example 3DCRT can also be performed with non-coplanar fields, it is not an exclusive feature of IMRT)

Patients with different prognoses (longer median survival for 4/10 patients) were included, although this has no influence, given the dosimetric purpose of the study.

Field width available for tomotherapy are 1, 2.5 and 5 cm. How did the authors use a 1.05 cm field width ?

The conclusion needs to be changed. The authors conducted a dosimetric study that showed that tomotherapy could give better coverage of the target and a decrease in the dose delivered to the organs at risk. Nothing more. It could be said that local control can be improved (and therefore that the entire theoretical demonstration has clinical significance) only after a randomized phase III study.

I thank the editor for the opportunity to review this manuscript.

Reviewer 3 Report

The study is very interesting and significant for daily clinical practice. 

Just few comments: 

  • in 2021 has been published the new WHO classification of brain tumors. You enrolled a patient with "oligoastrocytoma". This entity no onger exists in the new WHO classification. It could be useful to verify, if possible, molecular features of this tumor at diagnosis to correctly define this glioma.
  • In "materials and methods", you describe the simulation CT scan, and you wrote that the slices have a thickness of 3mm. It is admitted by classical guidelines, but why didn't you use the more pecise CT scan of 1mm thickness?
  • In "results" you compare the prognosis of 2 group of patients stratified on the PTV coverage. You said that " there was no significant difference between the two groups in terms of age, gender, ECOG, tumor location, tumor side of brain, operation type, WHO grade or chemotherapy administration".  However, there are other important features significant for the prognosis of these patients, in particular the molecular characterization (ex: IDH mutational status, 1p19q codelation, MGMT metilation, etc.). Can you include these features in your analisys?

Round 2

Reviewer 1 Report

Line 335: “From clinical results, better coverage could make patients with malignant glioma have better RFS and PFS. Taken together, TOMO planning might have the trend to achieve better disease local control.”

Please, revise the manuscript (including the title) smoothing your conclusions. Even if the statistical significance has been reached as regards the survival outcomes, the sample size is too small to propose any definitive conclusions. Underline throughout the entire manuscript that these results are very very preliminary (the number you used is typical of phase I trial, wich mainly investigates treatment's tolerability without any comparison with other treatments for evaluating survival outcomes, while the comparison you have done is more phase III-oriented trials, but without sufficiently large dataset) and need to be convalidated with a larger accrual. For example, just due to the small sample size, reported survival outcomes could be independent from tumor dose coverage. In my opinion, providing such analyses could be misleading and their weaknesses need to be duly highlighted in a dedicated limitations section.

Please, cite three additional references inherent with the IMRT and VMAT abilities and of other dosimetry-improving technical tricks:

  1. Ferini G, Valenti V, Puliafito I, Illari SI, Marchese VA, Borzì GR. Volumetric Modulated Arc Therapy Capabilities for Treating Lower-Extremity Skin Affected by Several Merkel Cell Carcinoma Nodules: When Technological Advances Effectively Achieve the Palliative Therapeutic Goal while Minimising the Risk of Potential Toxicities. Medicina (Kaunas). 2021 Dec 18;57(12):1379. doi: 10.3390/medicina57121379. PMID: 34946324; PMCID: PMC8703259.
  2. Ferini G, Tripoli A, Molino L, Cacciola A, Lillo S, Parisi S, Umina V, Illari SI, Marchese VA, Cravagno IR, Borzì GR, Valenti V. How Much Daily Image-guided Volumetric Modulated Arc Therapy Is Useful for Proctitis Prevention With Respect to Static Intensity Modulated Radiotherapy Supported by Topical Medications Among Localized Prostate Cancer Patients? Anticancer Res. 2021 Apr;41(4):2101-2110. doi: 10.21873/anticanres.14981. PMID: 33813420.
  3. Ferini G, Molino L, Tripoli A, Valenti V, Illari SI, Marchese VA, Cravagno IR, Borzi GR. Anatomical Predictors of Dosimetric Advantages for Deep-inspiration-breath-hold 3D-conformal Radiotherapy Among Women With Left Breast Cancer. Anticancer Res. 2021 Mar;41(3):1529-1538. doi: 10.21873/anticanres.14912. PMID: 33788746.

And this one about the potential role of liquid biopsy in glioma patients:

  1. Bonosi L, Ferini G, Giammalva GR, Benigno UE, Porzio M, Giovannini EA, Musso S, Gerardi RM, Brunasso L, Costanzo R, Paolini F, Graziano F, Scalia G, Umana GE, Di Bonaventura R, Sturiale CL, Iacopino DG, Maugeri R. Liquid Biopsy in Diagnosis and Prognosis of High-Grade Gliomas; State-of-the-Art and Literature Review. Life (Basel). 2022 Mar 11;12(3):407. doi: 10.3390/life12030407. PMID: 35330158; PMCID: PMC8950809.

Author Response

Response to Reviewer 1 Comments:

  1. Line 335: “From clinical results, better coverage could make patients with malignant glioma have better RFS and PFS. Taken together, TOMO planning might have the trend to achieve better disease local control.” Please, revise the manuscript (including the title) smoothing your conclusions. Even if the statistical significance has been reached as regards the survival outcomes, the sample size is too small to propose any definitive conclusions. Underline throughout the entire manuscript that these results are very preliminary (the number you used is typical of phase I trial, wich mainly investigates treatment's tolerability without any comparison with other treatments for evaluating survival outcomes, while the comparison you have done is more phase III-oriented trials, but without sufficiently large dataset) and need to be convalidated with a larger accrual. For example, just due to the small sample size, reported survival outcomes could be independent from tumor dose coverage. In my opinion, providing such analyses could be misleading and their weaknesses need to be duly highlighted in a dedicated limitations section.

Authors’ Response: Thanks for your comment and suggestion. I apologize that I do not present our entire theoretical demonstration clearly in our manuscript. I agree as your statement that our study is not a randomized-control phase III trial so I should not conclude directly. However, the manuscript in Line 335 had be suggested by Reviewer 2. Thus, I try to explain smoothly in my conclusion “Taken together, TOMO planning might have the trend to improve disease local control and this conclusion should be proved after a randomized phase III study in the future.” in the end of the conclusion.

  1. Please, cite three additional references inherent with the IMRT and VMAT abilities and of other dosimetry-improving technical tricks:

  1. Ferini G, Valenti V, Puliafito I, Illari SI, Marchese VA, Borzì GR. Volumetric Modulated Arc Therapy Capabilities for Treating Lower-Extremity Skin Affected by Several Merkel Cell Carcinoma Nodules: When Technological Advances Effectively Achieve the Palliative Therapeutic Goal while Minimising the Risk of Potential Toxicities. Medicina (Kaunas). 2021 Dec 18;57(12):1379. doi: 10.3390/medicina57121379. PMID: 34946324; PMCID: PMC8703259.→Modern RT technology advances effectively through VMAT to achieve nodules regression of Merkel cell carcinoma and prevent those patients' skin toxicity [41].

  1. Ferini G, Tripoli A, Molino L, Cacciola A, Lillo S, Parisi S, Umina V, Illari SI, Marchese VA, Cravagno IR, Borzì GR, Valenti V. How Much Daily Image-guided Volumetric Modulated Arc Therapy Is Useful for Proctitis Prevention With Respect to Static Intensity Modulated Radiotherapy Supported by Topical Medications Among Localized Prostate Cancer Patients? Anticancer Res. 2021 Apr;41(4):2101-2110. doi: 10.21873/anticanres.14981. PMID: 33813420.→A retrospective study [39] of prostate cancer patients analyzed rectal toxicity and proved that VMAT was more effective in decreasing proctitis than IMRT adding topical medications.

  1. Ferini G, Molino L, Tripoli A, Valenti V, Illari SI, Marchese VA, Cravagno IR, Borzi GR. Anatomical Predictors of Dosimetric Advantages for Deep-inspiration-breath-hold 3D-conformal Radiotherapy Among Women With Left Breast Cancer. Anticancer Res. 2021 Mar;41(3):1529-1538. doi: 10.21873/anticanres.14912. PMID: 33788746. →The pulmonary and cardiac radiation dose of left breast cancer can be reduced significantly via the deep-inspiration-breath-hold technique RT [40] and

And this one about the potential role of liquid biopsy in glioma patients:

  1. Bonosi L, Ferini G, Giammalva GR, Benigno UE, Porzio M, Giovannini EA, Musso S, Gerardi RM, Brunasso L, Costanzo R, Paolini F, Graziano F, Scalia G, Umana GE, Di Bonaventura R, Sturiale CL, Iacopino DG, Maugeri R. Liquid Biopsy in Diagnosis and Prognosis of High-Grade Gliomas; State-of-the-Art and Literature Review. Life (Basel). 2022 Mar 11;12(3):407. doi: 10.3390/life12030407. PMID: 35330158; PMCID: PMC8950809.A recent review article [34] has allowed an innovative tool, next-generation sequencing (NGS), to use liquid biopsy of glioma in the prediction of disease prognosis determining neural stem-like cells combine with different molecular markers to develop malignant glioma.

Authors’ Response: Thank you for your suggestion. I have cited those articles in the Discussion with red-marker words.
